# Walking and rolling of crystals induced thermally by phase transition

Takuya Taniguchi[1], Haruki Sugiyama[2], Hidehiro Uekusa[2], Motoo Shiro[3], Toru Asahi[1,3] & Hideko Koshima[3]

The mechanical motion of materials has been increasingly explored in terms of bending and expansion/contraction. However, the locomotion of materials has been limited. Here, we report walking and rolling locomotion of chiral azobenzene crystals, induced thermally by a reversible single-crystal-to-single-crystal phase transition. Long plate-like crystals with thickness gradient in the longitudinal direction walk slowly, like an inchworm, by repeated bending and straightening under heating and cooling cycles near the transition temperature. Furthermore, thinner, longer plate-like crystals with width gradient roll much faster by tilted bending and then flipping under only one process of heating or cooling. The length of the crystal is shortened above the transition temperature, which induces bending due to the temperature gradient to the thickness direction. The bending motion is necessarily converted to the walking and rolling locomotion due to the unsymmetrical shape of the crystal. This finding of the crystal locomotion can lead to a field of crystal robotics.

[1] Department of Advanced Science and Engineering, Graduate School of Advanced Science and Engineering, Waseda University, 3-4-1 Okubo, Shinjuku-ku, Tokyo 169–8555, Japan. [2] Department of Chemistry and Materials Science, Tokyo Institute of Technology, Ookayama 2-12-1, Meguro-ku, Tokyo 152-8551, Japan. [3] Research Organization for Nano & Life Innovation, Waseda University, 513 Wasedatsurumaki-cho, Shinjuku-ku, Tokyo 162-0041, Japan. Correspondence and requests for materials should be addressed to H.K. (email: hkoshima@aoni.waseda.jp)

Materials that can respond to external stimuli, such as light, heat, electricity, humidity, pH, and concentration gradients, have attracted much attention in chemistry, materials science, and engineering fields. Such materials are important for future applications in sensors, switches, actuators, artificial muscles, and soft robots. Many polymer and gel materials exhibit mechanical motions in response to external stimuli[1,2].

Recently, mechanical motions of molecular crystals have been increasingly reported. Solid-state photochemical reactions can cause molecular crystals to bend[3], twist[4,5], rotate[6], and jump[7,8]. Thermal single-crystal-to-single-crystal (SCSC) phase transitions also have caused crystal bending[9], expansion/contraction[10,11], and jumping[12]. We have reported the photomechanical bending of crystals of azobenzene[13–15], furylfulgide[16], salicylideneaniline[17,18], and anthracene[19] derivatives. Mechanical motions of molecular crystals have been reviewed in several publications[20–24].

However, the locomotion of materials has been limited. A few polymers and gels have shown locomotive features[25–27]. One molecular crystal has been reported to show slow crawling[28] due to repeated crystallization and melting under light irradiation. Despite present limitations, molecular crystals are expected to play important roles as locomotive materials and are being researched especially in the field of soft robotics[29].

We have recently reported that the chiral azobenzene derivative N-[[4-p-dimethylaminophenylazo]benzoyl]-1-phenylethylamine [trans-(S)-1] shown in Fig. 1 is crystallized to form two polymorphs—an α phase (space group: $P2_12_12_1$) and a β phase (space group: $P2_1$)—and thin, plate-like crystals of the β phase bend with twisting under ultraviolet (UV) light irradiation[15]. The β-phase crystal exhibits a reversible SCSC phase transition to a γ phase at 145 °C.

Herein, we report that the β crystals walk slowly, like an inchworm, by repeating its shape change when alternately heated and cooled near the transition temperature. More surprisingly, thin, long crystals roll very fast by repeated bending and then flipping under only one process of heating or cooling. The driving force of the both directional locomotion is generated from the unsymmetrical shape of the crystal. This walking and rolling crystal can be a beginning to the research and development of crystal robotics.

## Results

**Phase transition**. Differential scanning calorimetry (DSC) measurements of β crystals of trans-(S)-1 were performed in the temperature range of 60–160 °C at a rate of 10 °C min$^{-1}$ with heating and then cooling (Fig. 2a). On heating, the DSC curve showed an endothermic peak at 145.2 °C. On subsequent cooling, the DSC curve had an exothermic peak at 143.0 °C, showing the small thermal hysteresis of 2.2 °C. The enthalpies at these endothermic and exothermic peaks were small, 1.31 and −1.29 kJ mol$^{-1}$, respectively. The transition peak was also observed even after melt and crystallization (Supplementary Fig. 1).

Powder X-ray diffraction (XRD) profiles of β crystals of trans-(S)-1 changed between 140 and 150 °C on heating, and then returned to the initial pattern between 150 and 140 °C on cooling, showing that the reversible phase transition proceeded through a crystal-to-crystal process (Supplementary Fig. 2). Here, the crystal structure above the transition temperature is referred to as the γ phase.

**Crystal structures**. X-ray crystallographic analyses of trans-(S)-1 crystal were successfully performed at −100, 20, 100, 125, and 160 °C, confirming the SCSC phase transition from the β to γ crystal (Fig. 2b–f and Supplementary Table 1). Crystal structures at all temperatures belonged to the same space group, $P2_1$. The

**Fig. 1** Molecular structure of chiral azobenzene trans-(S)-**1**

lattice constants changed discontinuously at 160 °C due to the phase transition to the γ phase. The lengths of a and b axes shortened, by −4.2% and −0.30%, respectively, between 125 and 160 °C; that of the c axis elongated by +7.0% (Fig. 2b–d).

Crystal structures at 20, 100, 125, and 160 °C were solved as disordered structures, although there was no disorder at −100 °C (Fig. 2g). Here, we define the molecular conformation at −100 °C as conformer A and the other conformation as conformer B. Occupancy of conformer A decreased with temperature rise, and that of B increased, to give 0.62:0.38 at 125 °C in the β crystal, and 0.44:0.56 at 160 °C in the γ crystal (Supplementary Table 2).

These two conformers at 125 and 160 °C are illustrated to compare the molecular conformations (Fig. 2i, j). The disorder was caused by a pedal motion of the azobenzene moiety. The dihedral angle between the azobenzene plane and the phenyl ring of the side chain, 81.03°, of conformer A at 125 °C decreased considerably, to 56.17° at 160 °C due to the phase transition (Fig. 2h and Supplementary Table 2). The phenyl ring of the side chain in conformer A rotated by 25° (Fig. 2i, j). In contrast, the dihedral angles 87.93° (at 125 °C) and 82.88° (at 160 °C) of conformer B did not change significantly before and after the phase transition.

In the β-phase crystal, weak intermolecular NH---O=C hydrogen bond chains form between the amido groups, with distances of 2.603 and 2.728 Å for the A and B conformers, respectively, along the b axis, which is the longitudinal direction of the crystal (Fig. 3c, d, f). After the transition to the γ phase, the phenyl ring of the conformer A becomes almost perpendicular to the b axis and parallel to the c axis, leading to a slight shortening of the hydrogen bond distances (2.596 and 2.664 Å) for conformers A and B (Fig. 3d). This conformational change allows the molecules to push away surrounding molecules along the c axis and to condense along the b axis (Fig. 3c). However, the changes in molecular conformation and the packing arrangement are very small, which lead to the reversible SCSC phase transition with the small thermal hysteresis, 2 °C.

**Crystal shape changes**. When a piece of the plate-like β crystal (length 532 μm × width 133 μm × thickness 51.5 μm) was heated and then cooled on a conventional glass plate in the temperature range of 120–160 °C at a rate of 20 °C min$^{-1}$ under a polarized optical microscope, the crystal shape changed with no fracture due to the SCSC phase transition (Fig. 3). When viewed from the (100) top face of the crystal on heating, the length along the b axis shrank (−0.56%), and the width along the c axis expanded (+4.5%; Fig. 3a and Supplementary Movie 1). The crystal returned reversibly to the initial shape on subsequent cooling. The shape change of the crystal was nearly consistent with the changes in the unit cell lengths of the b axis (−0.30%) and c axis (+7.0%) at the phase transition. When viewed from the (010) cross-section face, the width along the c axis expanded (+4.5%), and the thickness along the a axis contracted (−2.9%) on heating and then returned to the initial shape on cooling (Fig. 3b and Supplementary Movie 2).

When viewed from the (001) side face on heating, we found that the plate-like crystal bent slightly along the b axis, and then became straight, with slight shortening (−0.56%) of the length

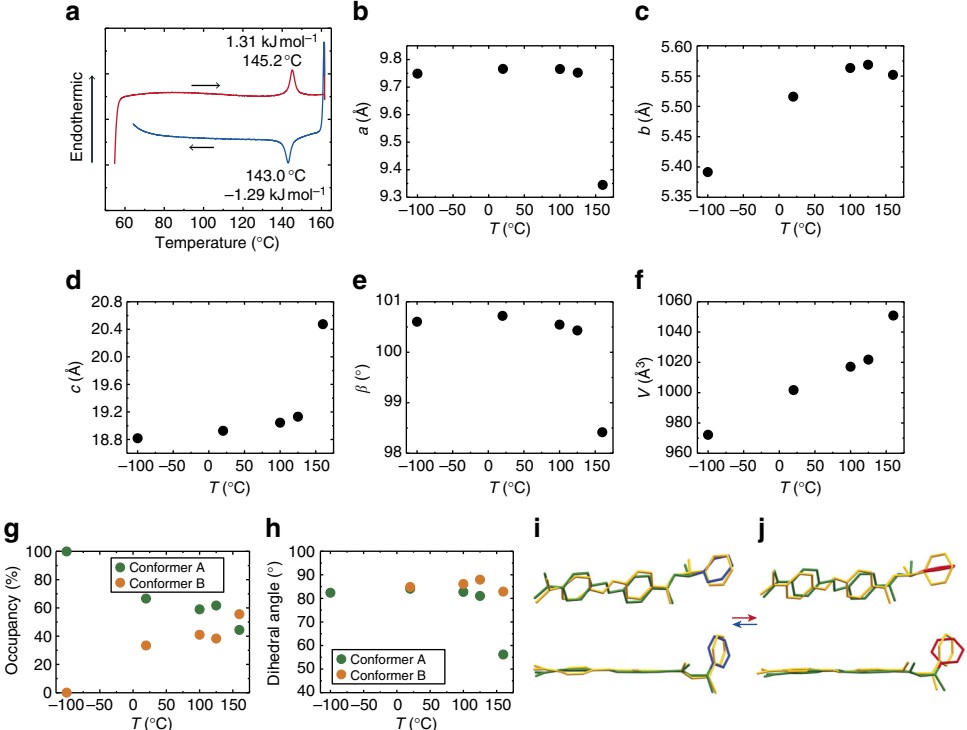

**Fig. 2** Characterization of the single-crystal-to-single-crystal phase transition. **a** Differential scanning calorimetry (DSC) curve measured in the temperature range of 60–160 °C at a rate of 10 °C min⁻¹ for heating and subsequent cooling. Red and blue lines represent heating and cooling, respectively. **b**–**f** Temperature dependence of lattice constants (−100, 20, 100, 125, and 160 °C); lengths of (**b**) *a*, (**c**) *b*, (**d**) *c* axes, (**e**) angle *β*, and (**f**) volume *V*. **g** Occupancies of two conformers A and B. **h** Dihedral angles between azobenzene plane and the phenyl ring of the side chain in each conformer. **i**, **j** Conformers A (green) and B (yellow) at (**i**) 125 and (**j**) 160 °C. The phenyl ring of the side chain of conformer A is shown in blue and red at 125 and 160 °C, respectively. Hydrogen atoms are omitted for clarity

along the *b* axis (Fig. 3e and Supplementary Movie 3). On subsequent cooling, the crystal bent again, and then returned to the initial shape.

This bending motion is caused most probably by a temperature gradient to the thickness direction of the crystal. On heating, the phase transition, from β to the γ phase, begins from the lower surface of the plate-like β crystal, which contacts the glass plate; thus, the crystal shrinks near the lower part, along the longitudinal axis, inducing the bending motion, as understood from the molecular arrangement change on the (001) side face (Fig. 3f). Then, the crystal becomes straight when the phase transition to the γ phase proceeds through the whole crystal. On subsequent cooling, the reverse phase transition, from γ to β phase, begins from the upper surface due to cooling by the surrounding air, and the crystal bends due to the elongation of the length. Then, the crystal returns to the initial straight shape when the transition to the β phase proceeds throughout the whole crystal.

A longer, thinner crystal (2100 × 50 × 25 μm³), the left edge of which was fixed with glue, bent more largely on heating and subsequent cooling, to reach a maximum bending angle, θ = 15° (Supplementary Fig. 4a, b and Supplementary Movie 4). This reversible bending was observed over 200 cycles of alternating heating and cooling (Supplementary Fig. 4c).

**Inchworm-like walking**. When a long plate-like crystal (length 8.5 mm, width 200 μm, one end thickness 190 μm, and other end thickness 55 μm) with a thickness gradient in the longitudinal direction was repeatedly heated and cooled between 139 and 154 °C at speeds of 0.6 and 1.0 °C s⁻¹, respectively, on a silanized glass, surprisingly, the crystal walked slowly to the right along the long

axis (*b* axis), like an inchworm, with repeated slight bending and straightening (Fig. 4a and Supplementary Movie 5). The walking distance reached 1.5 mm after 30 min (walking speed 3.0 mm h⁻¹). The average stroke on heating and cooling, estimated from the time profiles (Supplementary Fig. 5) extracted from Supplementary Movie 5, was 34 μm, which corresponds to 0.4% of the crystal length.

To further examine the mechanism of the inchworm-like walk, the surface temperature distribution of the crystal was monitored with an infrared (IR) thermography camera from the top of the crystal, and the crystal motion was observed simultaneously with a digital optical microscope from the side; the setup and the combined video are shown in Fig. 4b and Supplementary Movie 6, respectively. The speeds of heating and cooling between 132 and 150 °C were slowed to 0.2 and 0.4 °C s⁻¹, respectively, for simultaneous observation. Figure 4c and d show the snapshots of the crystal motion and the surface temperature distribution. The temperatures measured at the selected points were displayed on the crystal (left, middle, and right) and the silanized glass with cross marks (Fig. 4d).

On heating, the crystal remained straight until the surface temperature of the glass plate reached 140 °C (0 s, 45 s, Fig. 4c, d). Then, the crystal bent gradually with fluctuation (45–98 s) and the thicker left edge tended to move slightly (47 μm) to the right during this bending, as shown in the enlarged picture (80 s, Fig. 4c). At this time, the surface temperature at the middle became lower (142 °C) than those of the left (144 °C) and right (146 °C) sides due to rising up from the glass surface via bending (80 s, Fig. 4d, e), and then the temperature difference increased until the surface temperature at the middle reached 145 °C at almost 98 s (Fig. 4e).

The bending ended suddenly at 98 s, and simultaneously the right edge hopped up, like a spring (98 s, Fig. 4c), keeping the lifted shape until 105 s. At this time, the surface temperature of the right edge decreased, to 144 °C, while that of left and middle became 148–149 °C (98 s, Fig. 4d, e), and then the glass surface reached its maximum temperature (150 °C) on further heating (105 s, Fig. 4e). Next cooling finished the lifting up at the right edge at 118 s, and then crystal bending began again (120 s, Fig. 4c), decreasing the surface temperature rapidly in the middle, to ~141 °C (120 s, Fig. 4d, e). Then, the crystal bending decreased gradually with the decrease in the crystal surface temperature, and finally returned to the initial straight shape with movement (42 μm) of the right edge to the right (150 s, Fig. 4c).

The walking speed was estimated to be 1.0 mm h$^{-1}$ at the slowed heating (0.2 °C s$^{-1}$) and cooling (0.4 °C s$^{-1}$) rates, which was three times slower than that (3.0 mm h$^{-1}$) at the heating (0.6 °C s$^{-1}$) and cooling (1.0 °C s$^{-1}$) rates in Fig. 4a, showing that the walking speed is almost proportional to the heating and cooling rates. On a glass without surface treatment, the same crystal moved 1.1 mm in 30 min (walking speed 2.2 mm h$^{-1}$) to the right direction at the same heating and cooling rate in Fig. 4a (Supplementary Fig. 6), which was slower than the speed 3.0 mm h$^{-1}$ on a silanized glass.

To check the relationship between the crystal shape and the motion, we observed walking motion of a shorter plate-like crystal (length 4.1 mm) with a thickness gradient, of which length and thickness were around half of the crystal (length 8.5 mm) in Fig. 4 (Supplementary Fig. 7). When the shorter crystal was repeatedly heated and cooled on a silanized glass at the same temperature rate as in Fig. 4a, the crystal moved 0.89 mm in 30 min (walking speed 1.8 mm h$^{-1}$), giving around half speed of the longer crystal (walking speed 3.0 mm h$^{-1}$).

**Fast rolling locomotion.** More surprisingly, the crystal rolled very fast under only one process of heating or cooling. When a long, thin plate-like crystal (length 9.6 mm × left width 115 μm and right width 210 μm × thickness 70 μm) with a width gradient was heated on a silanized glass from 134 to 147 °C (at 1 °C s$^{-1}$), the crystal began to bend with fluctuation at ~137 °C. Then, the crystal rolled with repeating flip at 141 °C, and finally stopped at ~142 °C (Fig. 5a and Supplementary Movie 7). The locomotion distance reached 3.7 mm in only 0.8 s; the speed was thus 4.6 mm s$^{-1}$. This rolling locomotion was much faster than inchworm-like walking.

In one process of cooling of the same crystal, similar rolling was observed as easily understood from the slow-motion video

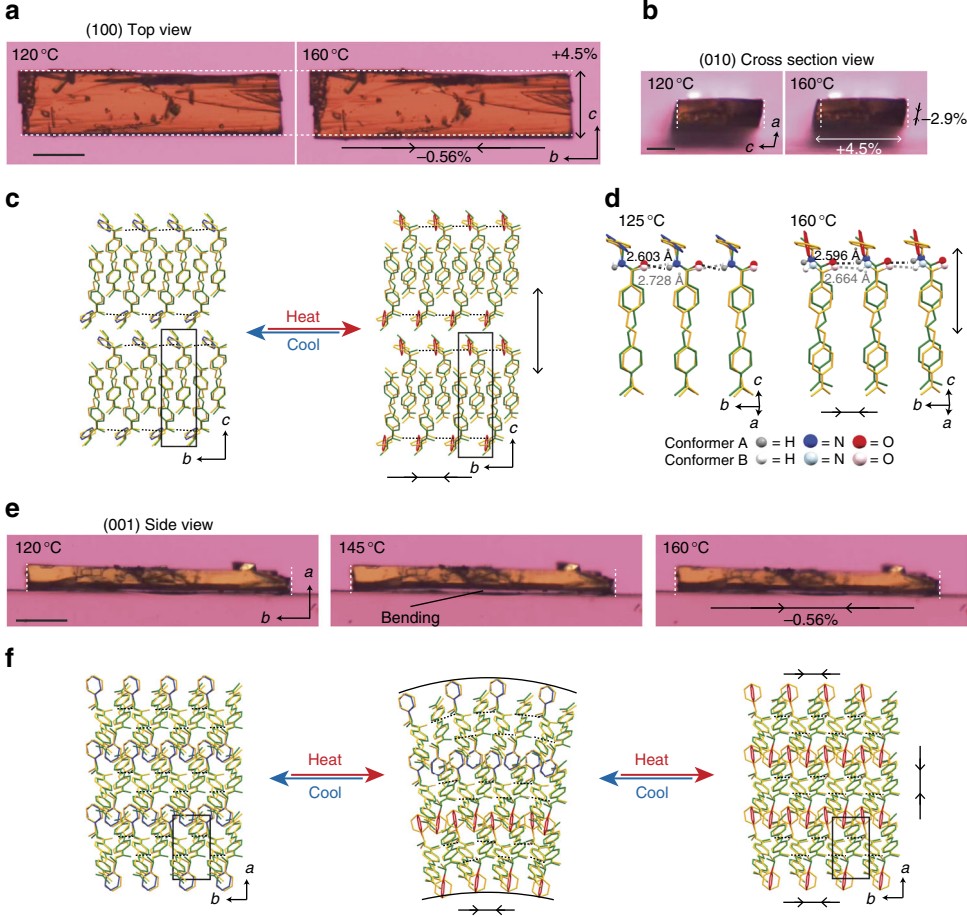

**Fig. 3** Changes of crystal shape and molecular arrangement under heating and cooling. **a** (100) Top view before and after phase transition (Supplementary Movie 1). The scale bar is 100 μm. **b** (010) cross-section view (Supplementary Movie 2). The scale bar is 50 μm. **c** Molecular packing on (100) face. **d** NH---O=C hydrogen bond chains along the b axis, which are derived from the two-fold helical molecular packing on the (010) face along the b axis (see Supplementary Fig. 3). **e** (001) side view (Supplementary Movie 3). The scale bar is 100 μm. Bending motion along the b axis was observed on heating and then cooling. In each panel, dotted lines in the pictures (drawn in white) indicate the initial size of the β crystal. **f** Molecular packing on (001) face. In packing diagrams, dotted lines (drawn in black) represent NH---O=C intermolecular hydrogen bonds along the b axis. Hydrogen atoms are omitted for clarity

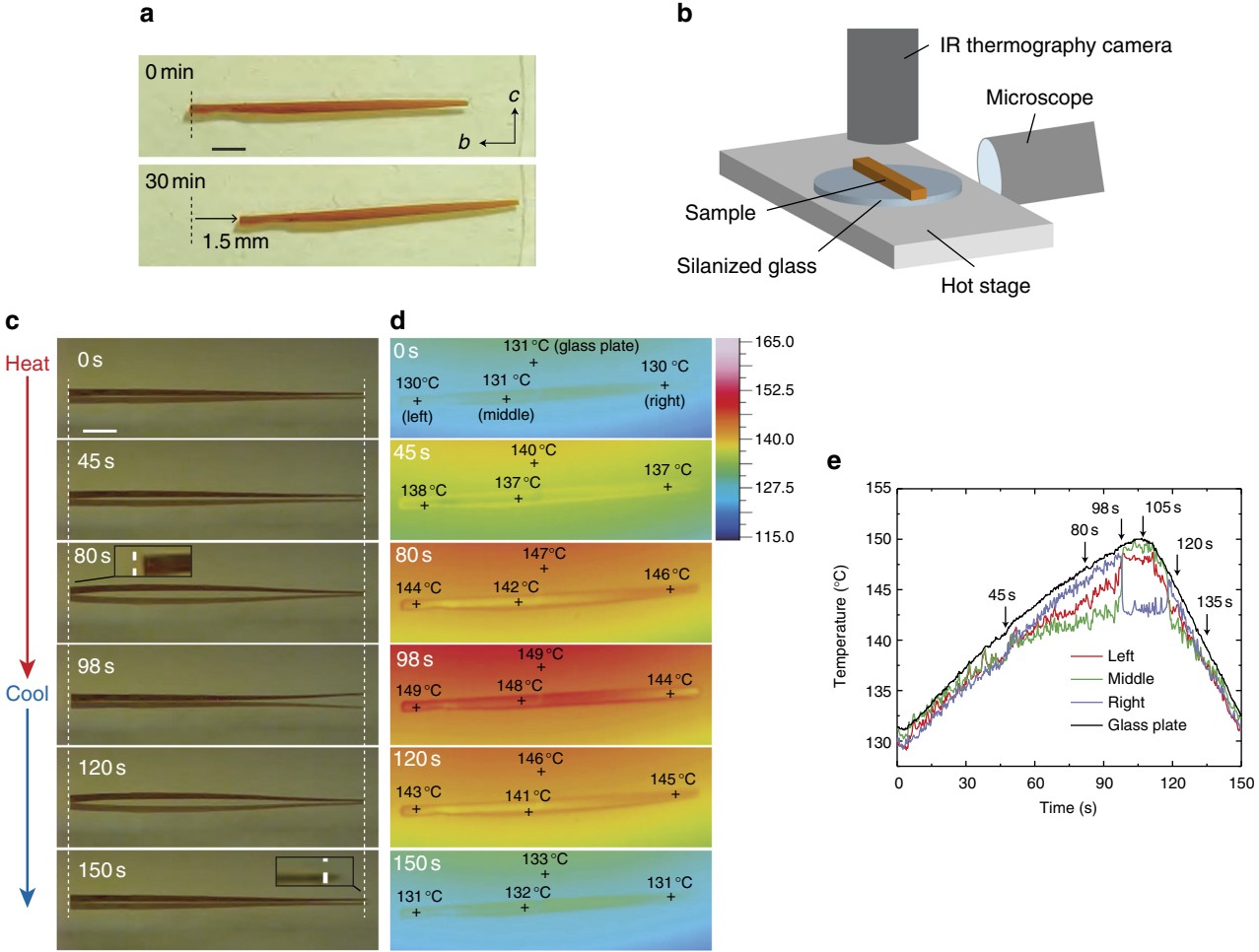

**Fig. 4** Inchworm-like walking of a long plate-like crystal with a thickness gradient. **a** Inchworm walking by repeated heating and cooling cycles (Supplementary Movie 5). The scale bar is 1 mm. **b** Experimental setup for simultaneous observation by microscope and infrared (IR) thermography camera. **c**, **d** Sequential snapshots during inchworm walk on heating and cooling, by optical microscope (**c**) and IR thermography (**d**) (Supplementary Movie 6). Dotted white lines in **c** indicate initial positions of the left and right edges. The scale bar is 1 mm. **e** Time dependence of surface temperatures at the selected points on the crystal and the glass plate, shown as a cross mark in **d**

(Supplementary Movie 8) and the snapshots (Fig. 5b). First, large bending occurred at ~140 °C; the bending was tilted, as detectable from the top view (28 ms, Fig. 5b). This tilted bending lost the balance to rotate the crystal by 90°, returning to almost straight shape (44 ms, Fig. 5b). Then, the first flip was completed by rotating another 90° due to momentum (52 ms, Fig. 5b). Subsequently, a smaller second bending with some tilt occurred (68 ms, Fig. 5b), and the crystal flipped again in a shorter time (28 ms) than the first flip (52 ms; at 80 ms, Fig. 5b). After the second flip (first rolling), the crystal continued to roll without noticeable bending, taking, on average, 38 ms for each roll and reaching a distance of 3.1 mm after 192 ms, showing that the rolling speed 16 mm s$^{-1}$ was approximately 20,000 times faster than that of the inchworm walk.

The slow-motion video (Supplementary Movie 9) taken from the slant upper side on cooling from 140 °C (at 1 °C s$^{-1}$) showed visually and more clearly the fast rolling locomotion. The first flip occurred in 30 ms, the second flip in 12 ms, and five consecutive rollings required, on average, 30 ms for each rolling (Fig. 5c). The moving distance reached 3.5 mm in 204 ms (rolling speed 17 mm s$^{-1}$).

When the right and left of the crystal were reversed on the silanized glass under the same cooling condition, the crystal rolled to the lower direction, which was opposite to the direction in Fig. 5b (Supplementary Fig. 8), suggesting that the direction of

rolling locomotion depended on the unsymmetrical shape of the crystal. On a glass without surface treatment, the crystal rolled repeatedly and moved 4.1 mm in 256 ms (rolling speed: 16 mm s$^{-1}$) (Supplementary Fig. 9), showing similar speed (16 mm s$^{-1}$) on a silanized glass.

The fast rolling locomotion was observed in several long, thin plate-like crystals of unsymmetrical shape (Supplementary Fig. 10). The crystals flipped to the direction where the angle of the crystal corner is obtuse angle. In contrast, shorter, thicker crystals tended to move much more slowly, without rolling; the example is shown in Supplementary Fig. 11 and Supplementary Movie 10. A correlation diagram between crystal shape (length and thickness) and locomotion with and without rolling is illustrated in Supplementary Fig. 12. The crystals of aspect ratio (thickness/length) 0.002–0.008 caused flipping. In contrast, the crystals of ratio 0.006–0.027 did not flip. Namely, shorter, thicker crystals cannot flip due to the small bending.

## Discussion

The schematic diagram of the possible mechanism of inchworm-like walk is shown in Fig. 6a. On heating, a plate-like β crystal with a thickness gradient bent due to the phase transition to the γ phase from the lower part (II, Fig. 6a). The larger bending in the

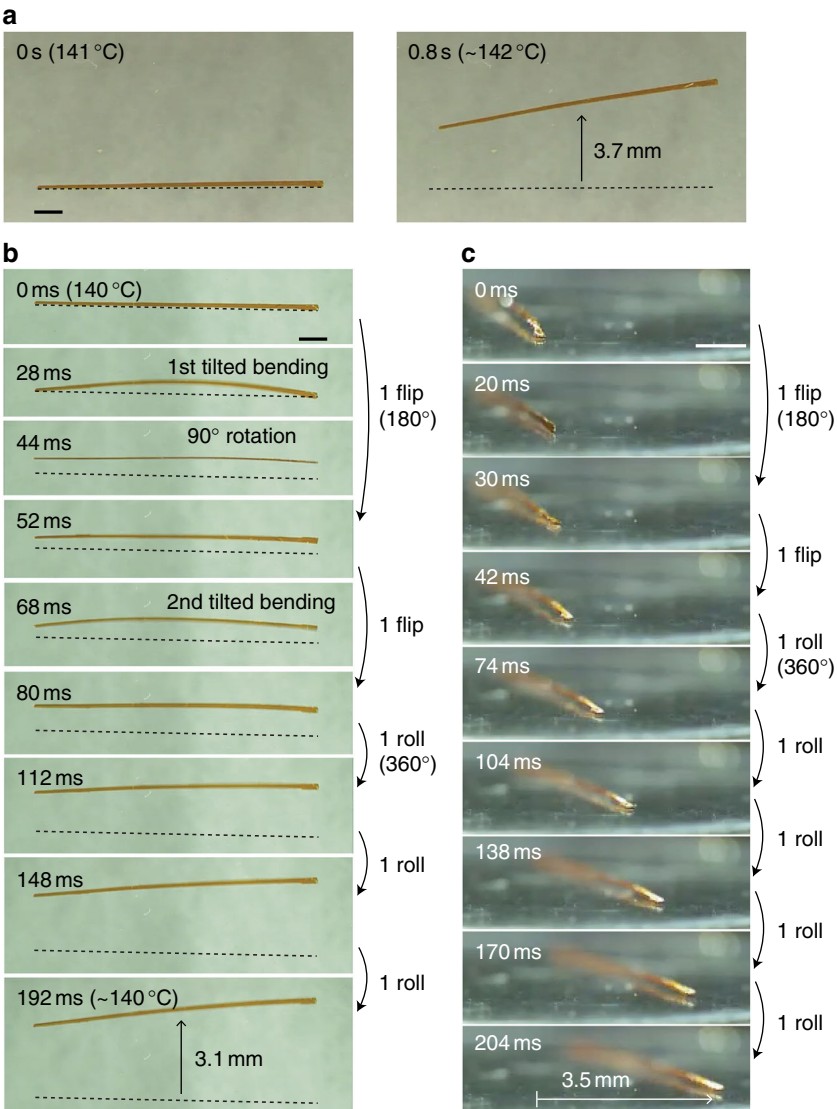

**Fig. 5** Fast rolling locomotion of a long, thin plate-like crystal. **a** Rolling of a long, thin plate-like crystal with a width gradient on heating (Supplementary Movie 7). **b** Sequential snapshots during the rolling locomotion on cooling (Supplementary Movie 8). Dotted lines indicate the initial position. **c** Slant upper side view of rolling locomotion of the same crystal on cooling (Supplementary Movie 9). The scale bar in **a–c** is 1 mm

thicker left part necessarily leads to movement to the right. When the phase transition to the γ phase proceeded throughout the whole crystal, the thicker left part became suddenly straight, and simultaneously the thinner right edge was lifted up like a spring, due to the relaxation of some strain generated by suddenly finishing bending (III, Fig. 6a).

On subsequent cooling, the thicker part started to bend again due to the phase transition from the γ to β phase near the crystal top surface due to cooling by the surrounding air (IV, Fig. 6a). When the whole crystal had finished the phase transition to the β phase, the thinner right edge moved largely to the right, by slipping on the silanized glass surface (V, Fig. 6a), because the thinner right part was lighter and thus the friction was less than in the thicker left part. In fact, the time profiles (Supplementary Fig. 5) support that the thicker left part moved largely during bending on heating, and then the thinner right edge moved largely on cooling, inducing the locomotion to the right, like an inchworm walk. The reason why the walking speed (2.2 mm h$^{-1}$) on a glass without surface treatment was slower than that (3.0 mm h$^{-1}$) on a silanized glass at the same heating and cooling rate is most probably due to the stronger interaction between the

hydrophilic glass surface and the crystal surface (Fig. 4a and Supplementary Fig. 6). Namely, the unsymmetrical crystal shape with the thickness gradient is the driving force of the inchworm-like walk.

A possible mechanism for the fast rolling locomotion is proposed (Fig. 6b), based on the observations in Fig. 5. When a long, thin plate-like crystal of the β phase is heated, the crystal bends with some tilt, due to the phase transition to the γ phase. The tilting autonomously occurs by the unsymmetrical crystal shape with the width gradient, to the direction where the angle of the crystal corner is obtuse angle, leading to loss of balance and the first flip; the incline of the center of gravity induces the driving force of the flipping. During the first flip, the whole crystal returns to the β phase due to cooling by the surrounding air. Then, the crystal turned inside out is again heated and bends with tilting due to the phase transition to the γ phase, to the direction where the angle of the crystal corner is acute angle likely due to the remaining momentum, leading to the second flip. When the whole crystal reaches the γ phase with further heating, the locomotion stops. Under cooling, the same explanation can be applied to the rolling locomotion (Fig. 6b).

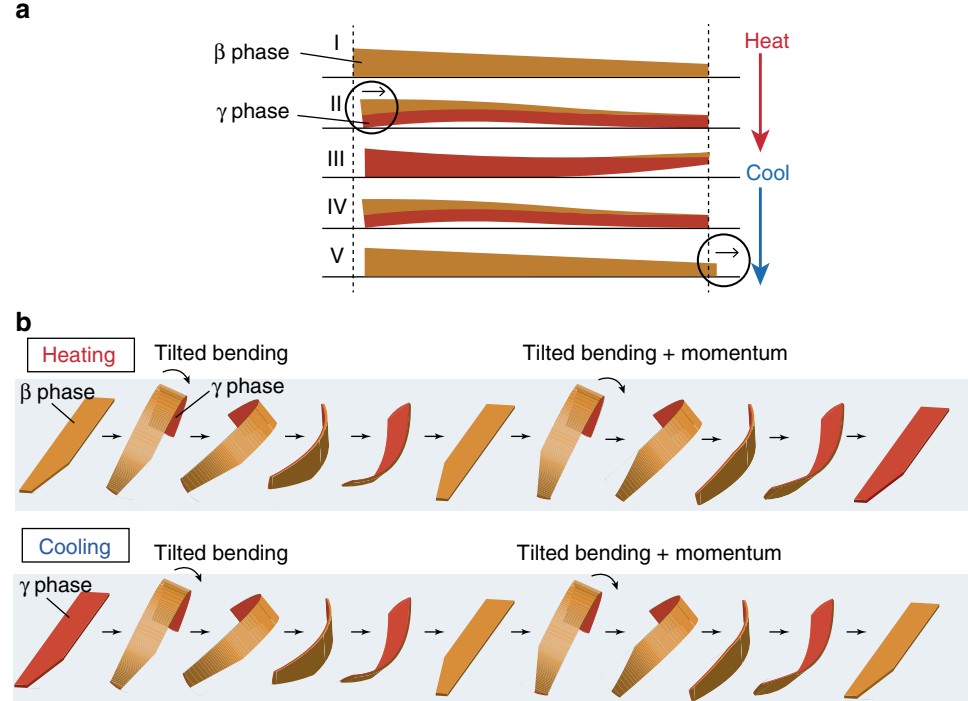

**Fig. 6** Schematic diagrams of the possible mechanisms of locomotion. **a** Inchworm-like walking of a long crystal with a thickness gradient by repeated heating and cooling cycles. Displacement to the right is indicated by the circle. **b** Rolling locomotion of a thin, long crystal with a width gradient under only one process of heating or cooling

Finally, some mechanical properties were evaluated; Young's modulus of the β-phase crystal, measured with a manual cantilever-bending test, showed 3.8 GPa (Supplementary Fig. 13), roughly comparable to several organic crystals[18,30,31]. The maximum strain and stress at the bending were estimated to be 0.48% (shrinkage) and 5.3 MPa, respectively (Supplementary Fig. 4b). The smooth following of the bending motion for the temperature change might be due to the small thermal hysteresis (2 °C) and the small enthalpy (1.3 kJ mol$^{-1}$) at the SCSC phase transition. During the bending and rolling, the plate-like crystal was elastic like a spring. The weak NH---O=C hydrogen bond chains (Fig. 3c, d, f and Supplementary Table 2) in the β and γ crystals should contribute to the spring-like behavior. In summary, it is concluded that the driving force of both the directional locomotions, the inchworm-like walking and the fast rolling, is generated from the unsymmetrical shape of the crystal.

## Methods

**Material preparation and characterization**. The $S$-enantiomer of chiral azobenzene, *trans*-($S$)-**1**, was prepared according to a previously published procedure[32,33]. Single crystals of the β form of *trans*-($S$)-**1** were recrystallized by slow evaporation from methanol or ethyl acetate at ambient temperature. DSC measurements of *trans*-($S$)-**1** samples were performed with a differential scanning calorimeter (DSC 8500, Perkin Elmer) by using ca. 3 mg of the samples and empty aluminum pan as reference, at a speed of 10 °C min$^{-1}$ for heating and cooling.

Powder XRD measurements were performed with a SmartLab system (Rigaku), equipped with monochromatic Cu–Kα radiation ($\lambda$ = 1.54187 Å) at 40 kV and 40 mA. A powdered sample of the β crystal of *trans*-($S$)-**1** was put on sample holder attached to a thermo-controller. Powder XRD data were collected at 25 °C and over the temperature range of 100–180 °C at intervals of 10 °C on heating and cooling. To allow for temperature stabilization, the interval was set at 1 min after reaching each temperature setting.

Single-crystal XRD data of β crystals of *trans*-($S$)-**1** were collected using an R-AXIS RAPID diffractometer (Rigaku) equipped with monochromatic Cu–Kα radiation ($\lambda$ = 1.54187 Å) at 50 kV and 100 mA. The temperature of the sample was regulated using a N$_2$ gas flow cryostat and calibrated with a thermocouple. Because the temperature fluctuated in a range of approximately ±3 °C, the average value was chosen for plotting the graphs. The sample was set in a glass capillary to

prevent oscillation by the N$_2$ gas flow and sublimation at high temperatures. The crystal structures were solved using a direct method with SHELXS2013[34] or SHELXD2013[34], and then refined on $F^2$ using the full-matrix least-squares method of SHELXL[34]. The calculations were performed using the Rigaku crystal structure software package[35] and a graphical interface, ShelXle[36]. Images of the crystal structures were rendered with POV-Ray[37] in Mercury[38].

**Observation of shape change and locomotion of crystals**. Shape change of a β crystal of *trans*-($S$)-**1**, at the phase transition, was observed on a glass plate using a polarized optical microscope (BX51, Olympus) with a microscopic thermo-controller (LMF-600S, Collet). The crystal was heated and then cooled in the temperature range of 120–160 °C at a rate of 20 °C min$^{-1}$. Locomotion of the crystal under the repeated cycles of heating and cooling was observed on a silanized glass (Hampton Research, HR3–239) under a digital high-speed microscope (VHX-500, Keyence). Temperature was regulated with a glass thermoplate (Tokai Hit). Temperature distribution during heating and cooling was measured by IR thermography (FSV-2000, Apiste) from the top of the crystal (resolution: 25 μm$^2$ per pixel). Crystal motion was recorded simultaneously with a microscope from the side of the crystal for synchronization with thermography measurements; the periods of heating and cooling were modified to allow for simultaneous observation.

**Data availability**. The crystallographic data have been deposited at the Cambridge Crystallographic Data Centre (CCDC) as CCDC 1556537-1556541 and can be obtained free of charge from the CCDC via www.ccdc.cam.ac.uk/getstructures.

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

## Acknowledgements

This study was financially supported by the JSPS Grant-in-Aid for Scientific Research B, Challenging Exploratory Research, and Research Fellowship for Young Scientists, and the Grant-in-Aid for Young Scientists (Early Bird) at Waseda Research Institute for Science and Engineering. We would like to thank Apiste Co. Ltd., for the use of IR thermography. T.T. and T.A. thank the Leading Graduate Program in Science and Engineering at Waseda University.

## Author contributions

T.T. and H.K. planned the experiments and wrote the paper, and T.T. performed all the experiments. H.S. and H.U. analyzed the crystal structure of γ phase. M.S. supported T.T. to analyze the crystal structures of β phase. H.K. and T.A. managed the project.

## Additional information

**Competing interests:** The authors declare no competing financial interests.

