## [Peer Review File · Nature Communications]

Reviewers' comments:

Reviewer #1 (Remarks to the Author):

The authors investigated the inchworm-like walking and flipping motions of azobenzene crystals on a silanized glass surface induced by a thermal phase transition of the crystal. Authors report that the thermal single-crystal-to-single-crystal (SCSC) phase transitions induces the bending of the crystal and then leading to the motion. Experiments have been well done and the results are explained well in the manuscript. The translational motions explained in this manuscript are very interesting and thus this manuscript is worth publishing in Nature Communications. However, I have several concerns to be addressed.

1) The authors used silanized glass, but I could not find the detail of the glass. Also, what happens if a glass without surface treatment is used?

2) As for the inchworm-like motion, is there any relationship between the crystal shape and the motion? Also, the I guess authors should have tried to optimize the condition of the rate of temperature change. I suggest authors to explain the relationship between the temperature rate and the motion (speed).

3) As for the running motion, the authors explain the relationship between the crystal size (dimensions) and the motion. In addition, I suppose the angle of the crystal corner is important for both the direction and rate of the flipping motion. The crystal flips to the direction where the angle of the crystal corner is obtuse angle.

Reviewer #2 (Remarks to the Author):

In this paper, the authors report 'walking' and 'running' locomotion of chiral azobenzene crystals, induced thermally by a reversible single-crystal-to-single-crystal phase transition. A long plate-like crystal with a thickness gradient in the longitudinal direction 'walked' slowly, like an inchworm, by repeated bending and straightening under heating and cooling cycles near the transition temperature, while thinner, longer plate-like crystals with width gradient 'ran' much faster by repeatedly tilted bending and then flipping under only one process of heating or cooling.

These are really interesting findings. Furthermore, the characterization of the crystals is fully performed and their movements are carefully and thoroughly analyzed. Basically, I recommend publication of this paper in Nature Communications if the authors could address to the following issues properly.

1) It seems to me that it is NOT appropriate to call the movement of the thinner, longer plate-like crystals with width gradient 'running' as it looks like 'rolling'.

2) The authors used silanized glasses as substrates for observation of motions of the azobenzene crystals. Are the silanized glasses necessary for the motion? What happens if the crystals are heated and cooled on other substrates such as glasses without treatment and metal plates?

3) The nomenclature 'crystbot' sounds strange as all the molecular crystals showing motions upon application of external stimuli are included in 'crystbot'.

Reply to comments

Reviewer 1:

Thank you for the good evaluation of our paper. We have revised the manuscript according to the comments.

Comment 1: The authors used silanized glass, but I could not find the detail of the glass. Also, what happens if a glass without surface treatment is used?

Reply 1: We used commercially available silanized glass (Hampton Research, HR3-239). The company name and the product code were added in the revised Method.

Page 19, line 8: “Locomotion of crystal under the repeated cycles of heating and cooling was observed on a silanized glass under a digital high-speed microscope (VHX-500, Keyence).” was changed to “Locomotion of crystal under the repeated cycles of heating and cooling was observed on a silanized glass (Hampton Research, HR3-239) under a digital high-speed microscope (VHX-500, Keyence).”

The walking speed was slowed on a glass without surface treatment, as shown in new Supplementary Fig. 6. In contrast, the running speed on a glass without surface treatment was almost similar to that on a silanized glass, as shown in new Supplementary Fig. 9. Following sentences were added in the revised Results.

Page 11, line -4: “On a glass without surface treatment, the same crystal moved 1.1 mm in 30 min (walking speed 2.2 mm h^{-1}) to the right direction at the same heating and cooling rate in Fig. 3a (Supplementary Fig. 6), which was slower than the speed 3.0 mm h^{-1} on a silanized glass.”

Page 13, line -2: “On a glass without surface treatment, the crystal flipped repeatedly and moved 4.1 mm in 256 ms (running speed: 16 mm s^{-1}) (Supplementary Fig. 9), showing similar speed (16 mm s^{-1}) on a silanized glass.”

Further, the sentences were added in the Discussion.

Page 15, line 10: “The reason why the walking speed (2.2 mm h^{-1}) on a glass without surface treatment was slower than that (3.0 mm h^{-1}) on a silanized glass at the same heating and cooling rate, is most probably due to the stronger interaction between the hydrophilic glass surface and the crystal surface (Fig. 3a and Supplementary Fig. 6).”

Comment 2: As for the inchworm-like motion, is there any relationship between the crystal shape and the motion? Also, I guess authors should have tried to optimize the condition of the rate of temperature change. I suggest authors to explain the relationship between the temperature rate and the motion (speed).

Reply 2: We obtained the result that the walking speed of the shorter crystal was slower than the longer crystal, as shown in new Supplementary Fig. 7. A new paragraph was added in the revised Results.

Page 11, last line: “To check the relationship between the crystal shape and the motion, we observed walking motion of a shorter plate-like crystal (length 4.1 mm) with a thickness gradient, of which length and thickness were around half of the crystal (length 8.5 mm) in Fig. 3 (Supplementary Fig. 7). When the shorter crystal was repeatedly heated and cooled on a silanized glass at the same temperature rate in Fig. 3a, the crystal moved 0.89 mm in 30 min (walking speed 1.8 mm h^{-1}), giving around half speed of the longer crystal (walking speed 3.0 mm h^{-1}).”

We examined the relationship between the temperature rate and the motion, and obtained the result that the walking speed was almost proportional to the heating and cooling rate. A new sentence was added in the revised Results.

Page 11, line 10: “The walking speed was estimated to be 1.0 mm h^{-1} at the slowed heating (0.2°C s^{-1}) and cooling (0.4°C s^{-1}) rate, which was three times slower than that (3.0 mm h^{-1}) at the heating (0.6°C s^{-1}) and cooling (1.0°C s^{-1}) rate in Fig. 3a, showing that the walking speed is almost proportional to the heating and cooling rate.”

Comment 3: As for the running motion, the authors explain the relationship between the crystal size (dimensions) and the motion. In addition, I suppose the angle of the crystal corner is important for both the direction and rate of the flipping motion. The crystal flips to the direction where the angle of the crystal corner is obtuse angle.

Reply 3: Thank you for the good comments. We observed very carefully the flipping direction of several crystals in Fig. 4 and Supplementary Fig. 10, and consequently we came to agree to your comment “The crystal flips to the direction where the angle of the crystal corner is obtuse angle.” However, we could not clarify the relation between the angle of the crystal corner and the rate of the flipping motion. At least, the longer, wider crystal needs longer flipping time (Supplementary Fig. 10a) than the shorter, narrower crystals (Fig. 4 and Supplementary Fig. 10b,c). A sentence was added in the revised Results.

Page 14, line 3: “The crystals flipped to the direction where the angle of the crystal corner is obtuse angle.”

Further, the sentences in the Discussion were revised as follows.

Page 16, line 1: “The tilting occurs by the unsymmetrical crystal shape with the width gradient, leading to loss of balance and the first flip; the incline of the center of gravity induces the driving force of the flipping. During the first flip, the whole crystal returns to the β phase due to cooling by the surrounding air. Then, the crystal turned inside out is again heated and bends with tilting due to the phase transition to the γ phase, leading to the second flip.” were changed to “The tilting autonomously occurs by the unsymmetrical crystal shape with the width gradient, to the direction where the angle of the crystal corner is obtuse angle, leading to loss of balance and the first flip; the incline of the center of gravity induces the driving force of the flipping. During the first flip, the whole crystal returns to the β phase due to cooling by the surrounding air. Then, the crystal turned inside out is again heated and bends with tilting due to the phase transition to the γ phase, to the direction where the angle of the crystal corner is acute angle likely due to the remaining momentum, leading to the second flip.”

Reviewer 2:

Thank you for the high evaluation of our paper. We have revised the manuscript according to the comments.

Comment 1: It seems to me that it is NOT appropriate to call the movement of the thinner, longer plate-like crystals with width gradient ‘running’ as it looks like ‘rolling’.

Reply 1: As the reviewer pointed out, the movement of the thinner, longer plate-like crystals with width gradient is unquestionably ‘rolling’. The speed of this ‘rolling’ locomotion is very fast like ‘running’, which is very interesting. In contrast, another ‘walking’ movement of long plate-like crystals with thickness gradient

is very slow like an inchworm. In order to emphasize comparison with the very slow inchworm-like 'walking', we would like to use the word 'running' by all means, rather than 'rolling' in this paper. The words 'walking' and 'running' are appeal point of this paper.

Comment 2: The authors used silanized glasses as substrates for observation of motions of the azobenzene crystals. Are the silanized glasses necessary for the motion? What happens if the crystals are heated and cooled on other substrates such as glasses without treatment and metal plates?

Reply 2: The walking speed was slowed on a glass without surface treatment as shown in new Supplementary Fig. 6. In contrast, the running speed on a glass without surface treatment was almost similar to that on a silanized glass as shown in new Supplementary Fig. 9. We did not try to observe crystal motion on metal plates. Following sentences were added in the revised Results.

Page 11, line -4: “On a glass without surface treatment, the same crystal moved 1.1 mm in 30 min (walking speed 2.2 mm h^{-1}) to the right direction at the same heating and cooling rate in Fig. 3a (Supplementary Fig. 6), which was slower than the speed 3.0 mm h^{-1} on a silanized glass.”

Page 13, line -2: “On a glass without surface treatment, the crystal flipped repeatedly and moved 4.1 mm in 256 ms (running speed: 16 mm s^{-1}) (Supplementary Fig. 9), showing similar speed (16 mm s^{-1}) on a silanized glass.”

Further, the sentences were added in the Discussion.

Page 15, line 10: “The reason why the walking speed (2.2 mm h^{-1}) on a glass without surface treatment was slower than that (3.0 mm h^{-1}) on a silanized glass at the same heating and cooling rate, is most probably due to the stronger interaction between the hydrophilic glass surface and the crystal surface (Fig. 3a and Supplementary Fig. 6).”

Comment 3: The nomenclature ‘crystbot’ sounds strange as all the molecular crystals showing motions upon application of external stimuli are included in ‘crystbot’.

Reply 3: We deleted the nomenclature ‘crystbot’. The last two sentences in the Abstract were revised.

Page 2, last line: “We name the robot-like crystal ‘crystbot’. This finding will lead to a new field of crystal robotics ‘crystbot’.” were changed to “This finding enables to open a new field of crystal robotics.”

Reviewers' comments:

Reviewer #1 (Remarks to the Author):

Authors have made a serious effort to revise and improve the manuscript. All my comments have been addressed adequately. I recommend publication as is.

Reviewer #2 (Remarks to the Author):

I believe that the mode of motions of these materials should be described as precisely as possible. The walking and running are in the same mode of motions. If one increases the speed of walking, it will be recognized as running. However, in this paper, what the authors refer to 'running' is far from what they refer to 'walking'; the mode of motion is completely different between 'walking' and 'running' in this paper. I still believe that it is NOT appropriate to call the movement of the thinner, longer plate-like crystals with width gradient 'running'; it should be called 'rolling' as the authors themselves approved in their 'Reply 1'.

Reply to comments

Reviewer 1:

Thank you for your recommendation for publication of our paper.

Reviewer 2:

We have revised the manuscript according to the comment.

Comment: I believe that the mode of motions of these materials should be described as precisely as possible. The walking and running are in the same mode of motions. If one increases the speed of walking, it will be recognized as running. However, in this paper, what the authors refer to 'running' is far from what they refer to 'walking'; the mode of motion is completely different between 'walking' and 'running' in this paper. I still believe that it is NOT appropriate to call the movement of the thinner, longer plate-like crystals with width gradient 'running'; it should be called 'rolling' as the authors themselves approved in their 'Reply 1'.

Reply: According to the comment, we use 'roll' to describe the flipping locomotion for accurate expression at first in the Result section. This rolling locomotion was much faster than inchworm-like walking. In order to emphasize the much faster speed, we name the rolling locomotion 'running'.

Following sentences were modified.

Page 12, line 7: "More surprisingly, the crystal 'ran' very fast by flipping under only one process of heating or cooling." was changed to "**More surprisingly, the crystal rolled very fast under only one process of heating or cooling.**"

Page 12, line 10: "Then, the crystal ran with repeating flip at 141°C, and finally stopped at ~142°C (Fig. 4a and Supplementary Video S7)." was changed to "**Then, the crystal rolled with repeating flip at 141°C, and finally stopped at ~142°C (Fig. 4a and Supplementary Video S7).**"

Further, following sentences were added.

Page 12, line 12: "**This rolling locomotion was much faster than inchworm-like walking. In order to emphasize the much faster speed, we name the rolling locomotion 'running'.**"